# Regulation of microRNA Expression in Sleep Disorders in Patients with Epilepsy

**DOI:** 10.3390/ijms22147370

**Published:** 2021-07-09

**Authors:** Edyta Dziadkowiak, Justyna Chojdak-Łukasiewicz, Piotr Olejniczak, Bogusław Paradowski

**Affiliations:** 1Department of Neurology, Wroclaw Medical University, 50-367 Wroclaw, Poland; justyna.ch.lukasiewicz@gmail.com (J.C.-Ł.); bogusparad@poczta.onet.pl (B.P.); 2Department of Neurology, Louisiana State University Health Sciences Center in New Orleans, 1542 Tulane Avenue, New Orleans, LA 70112, USA; polejn@lsuhsc.edu

**Keywords:** sleep-related seizures, epilepsy, parasomnias, sleep disorders, microRNA

## Abstract

The effects of epilepsy on sleep and the activating effects of sleep on seizures are well documented in the literature. To date, many sleep-related and awake-associated epilepsy syndromes have been described. The relationship between sleep and epilepsy has led to the recognition of polysomnographic testing as an important diagnostic tool in the diagnosis of epilepsy. The authors analyzed the available medical database in search of other markers that assess correlations between epilepsy and sleep. Studies pointing to microRNAs, whose abnormal expression may be common to epilepsy and sleep disorders, are promising. In recent years, the role of microRNAs in the pathogenesis of epilepsy and sleep disorders has been increasingly emphasized. MicroRNAs are a family of single-stranded, non-coding, endogenous regulatory molecules formed from double-stranded precursors. They are typically composed of 21–23 nucleotides, and their main role involves post-transcriptional downregulation of expression of numerous genes. Learning more about the role of microRNAs in the pathogenesis of sleep disorder epilepsy may result in its use as a biomarker in these disorders and application in therapy.

## 1. Introduction

MicroRNAs (miRNAs) are a group of small non-coding RNAs that, in their mature form, regulate gene expression at the post-transcriptional level. A single miRNA molecule can simultaneously control the expression of hundreds of target genes. More than 1/3 of protein-coding genes in human cells are regulated by microRNAs. It is estimated that genes encoding microRNAs constitute 1–5% of all genes in humans and animals [1,2].

The formation of microRNAs consists of several steps. The first is transcription, which leads to the formation of a primary microRNA (pri-microRNA) transcript. The next is processing of the pri-microRNA, which results in the formation of pre-microRNA. Both steps occur in the cell nucleus. The pre-microRNA is then transferred to the cytoplasm, where it undergoes processes leading to the formation of a mature, functional miRNA molecule of ~20 nt in length.

Mature microRNAs are present in the cell as a part of complex ribonucleoprotein complexes, among which the RNA-induced silencing complex (RISC) plays a special role. The RNA component acts as a probe that allows RISC to attach to a complementary transcript or DNA fragment to be regulated. In most cases, Nucleotides 2–7 of the microRNA (the so-called “seed” region) are crucial for interaction between the microRNA and the target nucleic acid molecule. The influence of RISC on the translation process consists in: blocking initiation or elongation, forcing premature dissociation of ribosome from mRNA, and, probably, inducing degradation of the nascent polypeptide. Additionally, microRNAs may participate in chromatin reorganization and silencing of genes at the transcriptional level. A single miRNA can interact with up to several hundred target mRNAs, often in cooperation with other miRNAs, forming an extremely complex regulatory network. More than 60% of human protein-coding genes have at least one conserved miRNA binding site. Considering the existence of many non-conservative interaction sites, it can be speculated that the extent of regulation by microRNAs is even broader. Numerous scientific reports demonstrate that the multiplicity of modes of action and complexity of microRNA interaction networks can control a variety of biological processes: cell division, cell differentiation, apoptosis, angiogenesis, or oncogenesis. Therefore, uncontrolled quantitative and qualitative changes in mature microRNAs may contribute to the development of pathological conditions, including epilepsy. It has been shown that the expression of proteins involved in pathways regulating neuronal functions (neuronal morphology, synaptic plasticity, long-term synaptic plasticity, and long-term synaptic potentiation (LTP)) is regulated by microRNAs [3,4,5].

MicroRNAs whose regulation affects sleep homeostasis and circadian rhythm include microRNA-107, microRNA-124, micorRNA-125a-3p, microRNA-132, microRNA-182, microRNA-126, and microRNA-146a [6,7]. MicroRNA-124 regulates the development of neurons by controlling key differentiation factors such as REST (repressor element-1 silencing transcription factor) and PTBP1 (polypyrimidine tract-binding protein 1) as well as maturation through effects on CREB (CRE binding protein―cAMP response element). Furthermore, by inhibiting the expression of the transcription factor Sox9 (sex determining region Y-box 9), neurogenesis in adults is increased. There is also an association between microRNA-132 and long-term memory, which is associated with the presence of CRE sequences in the microRNA-132 promoter to the CREB transcription factor, whose phosphorylation is necessary for the improvement of long-term memory. This affects the regulation of the expression of glutamate receptors (NR2A, NR2B, and GluR1), elongation of dendrites, formation of dendritic spines and dendritic spikes, modification of the transcription of key LTP-regulated genes, and thus neuronal development and plasticity [8,9].

Sleep is the basic biological requirement of the body to maintain body homeostasis. During sleep, complex information processing occurs. Sleep structure consists of non-rapid eye movement sleep (NREM) and rapid eye movement sleep (REM). According to the American Academy of Sleep Medicine classification of sleep stages, during NREM sleep, there are three stages, designated N1 (shallowest sleep), N2, and N3 (deepest sleep). REM sleep is designated stage R. Wakefulness during the night is denoted as W (wake). During the night, NREM and REM sleep follow each other cyclically (N1–N2–N3–N2–REM), forming so-called sleep cycles. Each sleep cycle usually ends with a brief awakening and repeats 4–6 times throughout the night. In the first part of the night, deep sleep—N3 stage—predominates, while, in the second part of the night, after the completion of three sleep cycles, REM sleep—N2 stage—predominates [10,11,12,13].

At the molecular level, sleep and wakefulness phases are regulated by not only classical neurotransmitters such as norepinephrine, serotonin, acetylcholine, histamine, gamma-aminobutyric acid (GABA), and glutamic acid but also many other chemicals called sleep factors. These include adenosine, growth hormones prolactin, prostaglandin D2, certain cytokines, and peptides from the hypocretin group, which include orexins [14,15,16,17].

Sleep disorders can lead to behavioral and cognitive disturbances and even disintegration of physiological body processes. Literature data show that patients with epilepsy have significantly lower sleep efficiency, more N2 non-rapid eye movement (NREM) sleep, and less rapid eye movement (REM) sleep with prolonged latency to REM sleep in comparison to the general population [10].

The connection between sleep disturbances and epilepsy is very well documented. Both conditions have a bidirectional relationship. Poor sleep may trigger worse seizure control and worse seizure control may trigger sleep disturbances. Anti-epileptic drugs can have an influence on normal patterns of sleep and cause daytime sleepiness. Another problem is that some types of seizures may occur during sleep [18,19,20,21].

The primary method for assessing circadian rhythm sleep–wake disorders (CRSWD) is sleep diaries, while chronotype (morning–evening) scales may be used optionally. The basic biological method of assessing circadian rhythm is the evaluation of the rhythm of rest and activity, or actigraphy. This study is performed using small recorders that contain a motion and light sensor. In addition to actigraphy, an objective assessment of circadian rhythm can be made by measuring physiological processes, the variability of which is largely regulated by the biological clock. Such measurements include determinations of the daily rhythm of core body temperature, the onset of growth of melatonin secretion in the dark (dim light melatonin onset (DLMO)), and diurnal changes in the concentration of 6-sulfatoxymelatonin (aMT6S) in urine.

The authors question whether microRNA determination can be useful in the evaluation of sleep disorders in patients with epilepsy and in differential diagnosis [22,23,24].

## 2. Methods

The authors conducted a literature search focused on the topic of the relationship among sleep, epilepsy, and microRNA. The key search terms applied in PubMed via MEDLINE and Google Scholar were “sleep” and “sleep disorders” and “epilepsy” or “microRNA”. The reference lists from eligible publications were searched online for their relevance to the topic. Reviews and research studies, classified according to their relevance, were included.

## 3. The Effect of Epilepsy on Sleep

Epilepsy can disturb sleep, even during a seizure-free night. Reported abnormalities in sleep architecture in epilepsy patients include reduced total sleep time, prolonged sleep latency, shortened and disturbed REM sleep, and sleep fragmentation with frequent stage changes and awakenings. Scarlatelli-Lima et al. [25] found that, in patients with temporal lobe epilepsy accompanied by hippocampal sclerosis, processes related to NREM sleep, mainly deep NREM sleep, activate focal interictal epileptic discharges (IEDs) and IEDs recorded during sleep, particularly in REM stage, have higher localizing value.

Nocturnal seizures may occur immediately after falling asleep, immediately before waking, and shortly after waking. Any seizure can occur while asleep. However, there are some seizure conditions where the patient is more likely to experience nocturnal seizures, including: sleep-related epilepsy hypermotor (SHE), benign partial epilepsy with centrotemporal spikes (BECTS), Panayiotopoulos syndrome (PS), juvenile myoclonic epilepsy (JME), awakening epilepsy (AE), and Landau–Kleffner syndrome.

Awakening epilepsy (AE) is an age-related syndrome of idiopathic generalized epilepsy (IGE) characterized by generalized tonic-clonic seizures (GTCS) occurring predominantly on awakening (independent of the time of day) or at leisure time (almost at evening) [26]. Landau–Kleffner syndrome (LKS) is characterized by acquired aphasia or verbal auditory aphasia, with or without focal seizures, secondarily generalized tonic-clonic seizures, absences, or atonic seizures [27]. Landau–Kleffner syndrome and continuous spike wave in slow-wave sleep (CSWS) can be associated with the electrical status epilepticus in sleep (ESES) described as an electroencephalographic pattern showing significant activation of epileptiform discharges in sleep [28]. In animal models of status epilepticus, miRNA-23a was upregulated in the hippocampus after status epilepticus [29,30].

Moustafa et al. [31] studied 50 patients with epilepsy, including 15 patients with idiopathic generalized epilepsy (8 patients with juvenile myoclonic epilepsy, 5 patients with epilepsy with generalized tonic-clonic seizures on awakening, and 2 patients with childhood absence epilepsy), 15 patients with focal epilepsy (3 idiopathic and 12 cryptogenic), and 20 healthy controls. They showed that serum miRNA 194–5P and miRNA 106b can be used as potential non-invasive biomarkers in the evaluation of idiopathic generalized epilepsy.

## 4. Impact of microRNAs on Sleep-Related Epilepsy

### 4.1. Sleep-Related Seizures―Selected Epilepsy Syndromes

The sleep–wake cycle has an effect on epilepsy. Sleep-related hypermotor epilepsy (SHE) and Panayiotopoulos syndrome (PS) are two of the most frequently implicated epilepsies occurring during the sleep state [2]. Generalized epilepsy, such as juvenile myoclonic epilepsy (JME), is most often associated with awakening.

SHE occurs during the non-rapid eye movement (NREM) stage of sleep. The hypothesis proposed by Zupcic et al. [32] is that seizures and the development of epileptogenesis in SHE are a consequence of cholinergic dysfunction and decreased levels of microRNA-211, as opposed to NREM parasomnias, where there is a stable level of microRNA-211, preventing epileptogenesis despite the cholinergic system dysfunction.

BECTS is the most common childhood epilepsy syndrome, in which seizures occur almost exclusively in sleep at the transition between rapid eye movement (REM) and non-REM cycles [33]. The clinical features include oropharyngolaryngeal symptoms (OPLS) (present in 50% of patients), speech arrest (40% of patients), unilateral facial sensorimotor symptoms (30% of patients), and hypersalivation (30% of patients). Characteristic changes in electroencephalographic recordings are centrotemporal spikes (CTS) arising independently in the right and/or left hemispheres from a normal background activity. Typical and atypical BECTS are presumed to have a shared genetic etiology. The pathogenesis of BECTS has been linked to the Elongator Complex (also called PAXNEB)―in particular, Elp 4–6 maintain translational fidelity via regulation of tRNA modifications. It has been shown that there are active genes located inside chromosomes, as exemplified by the approximately 1 Mpz region of 11p13 which contains, in addition to stretches of intergenic DNA of approximately 300 kpz, a large number of genes whose expression is regulated by ubiquitination (RCN and PAXNEB). The PAXNEB gene encodes a protein involved in elongation and is a homolog of elongation protein 4 in *Saccharomyces cerevisiae* [34]. GRIN2A gene mutations are more commonly found in the atypical form of BECTS. GRIN2A encodes the GluN2A subunit of the NMDAR. The functional consequences of mutations in GRIN2A are altered Zn2+ binding and loss of Zn2+ inhibition, which plays a critical role in normal neuronal development, synaptic plasticity and memory [35,36,37,38,39].

Panayiotopoulos syndrome is defined as a benign, age-related epilepsy syndrome characterized by seizures with predominant autonomic symptoms and multifocal EEG changes predominantly in the occipital region. It has been hypothesized that a mutation in SCN1A, the gene encoding the α-subunit of the brain type I voltage-gated sodium channel Nav1.1, may cause susceptibility to the occurrence of PS. Voltage-isolated sodium channel Nav1.1 plays an important role in controlling neuronal excitability. MicroRNA-155 is believed to target Nav1.1 and may play a role in the seizure inhibitory effects of valproic acid. Silencing of Nav1.1 by microRNA is an important regulator of neuronal excitability in epilepsy [4].

Juvenile myoclonic epilepsy (JME) is characterized by myoclonic jerks 1–2 h after awakening. It is a heterogeneous syndrome with an autosomal dominant inheritance. Genetic analyses of families with JME have revealed mutations in numerous genes: EFHC1, ClCN2, KCNQ3, KCNMB3, GABRA1, and BRD2. Less documented for the pathogenesis of this epilepsy syndrome are mutations within KCNJ10 and CACNA1A. Seizures are sometimes provoked by fatigue, sleep deprivation, emotions, and alcohol abuse [40,41] (Table 1).

### 4.2. Sudden Unexpected Death in Epilepsy

Sudden unexpected death in epilepsy (SUDEP) is defined as “sudden, unexpected, witnessed or unwitnessed, non-traumatic, and non-drowning death in patients with epilepsy with or without evidence for a seizure, and excluding documented status epilepticus, in which postmortem examination does not reveal a structural or toxicological cause of death” [42,43]. SUDEP is responsible for a total of about 8–17% of the causes of death in epilepsy patients. In adults, the prevalence of SUDEP is estimated at approximately 1.2 cases per 1000 people per year [42]. The most probable, commonly known risk factors for SUDEP are early age of onset, male gender, long duration of the disease, generalized tonic-clonic seizures (GTCS), the underlying disease causing the epilepsy, polytherapy, and patient’s lack of cooperation in the treatment process. The structural brain abnormalities, abnormal neurological assessment and intellectual disability, and psychiatric comorbidities also predispose to SUDEP [44]. In addition, the risk of SUDEP increases when seizures occur during sleep and at night [42,45,46]. In 2015, Mostacci et al. found that patients with nocturnal frontal lobe epilepsy (the syndrome’s name has been changed from autosomal dominant nocturnal frontal lobe epilepsy to sleep-related hypermotor epilepsy) did not show a higher risk of SUDEP. They suggested the need for an additional risk factor for SUDEP, possibly the occurrence of GTCS [47]. SUDEP pathomechanisms may result from arrhythmias, paroxysmal cardiomyopathy, dysfunction of the autonomic nervous system, and seizure-related respiratory failure. Nashef et al. [43] indicated the possibility of a coexisting susceptibility to sudden cardiac death―independent of or related to the epilepsy―that becomes symptomatic in the presence of uncontrolled seizures. It is extremely important that there is a group of epilepsy patients with various types of cardiac repolarization abnormalities that also occur in the interictal period. The occurrence of these changes, regardless of epileptic seizures, may be associated with gene mutations (e.g., leading to long QT syndrome and catecholaminergic polymorphic ventricular tachycardia) and mutations of the sodium channel genes SCN1A, SCN5A, potassium KCNH2, etc., which may cause the clinical picture of the disease with seizures, epilepsy, and fatal arrhythmias. Genetically conditioned shortening of the QT complex may be associated with peri-paroxysmal tachyarrhythmia and an increased risk of SUDEP. The discovery of mutations in the KCNQ1 gene in laboratory animals is associated with their predisposition to the occurrence of long QT syndrome and epileptic seizures. Because long QT is associated with an increased risk of arrhythmias, it appears to be responsible for a certain percentage of SUDEP cases as well [45,48,49].

Scorza et al. [45] suggested, that modifications to the expression pattern of circulating miRNAs may be associated with abnormal underlying cardiovascular processes and may be identified and used as SUDEP biomarkers [45]. De Matteis et al. [50] conducted a study of patients with SUDEP compared with 10 autopsies of traumatic or asphyxia deaths. They analyzed the expression profiles of several miRNAs (miR-301a-3p, miR-194–5p, miR-30b-5p, mIR-342-5p, and miR-4446-3p) from the plasma and temporal lobe and identified upregulation of miR-301a-3p in the plasma (2.3-fold) and hippocampus (3.2-fold) for SUDEP vs. controls [42,50]. Pansani et al. [51], in animal models of epilepsy, found in the group of rats with epilepsy and five GTCS increased in microRNA-21 and decreased in microRNA-320 expression compared to the group of rats without epilepsy and the group of rats with epilepsy and ten GTCS. Therefore, seizures impair cardiac function and morphology, probably through microRNA modulation [51].

A new class of specific circulating miRNAs has been identified as potential biomarkers of cardiovascular disorders, therefore there is a reasonable focus that the same molecules could also be useful in the investigation of SUDEP [45]. However, most authors believe that this potential biomarker still needs to be confirmed with additional cases.

## 5. The Classifications of Sleep Disorders

The most widely used classification of sleep disturbances is the third edition of the International Classification of Sleep Disorders (ICSD), which works in parallel with the fifth edition of the Diagnostic and Statistical Manual of Mental Disorders (DSM-5). The ICSD includes seven major categories of sleep disturbances: insomnia, sleep-related breathing disorders, central disorders of hypersomnolence, circadian rhythm sleep–wake disorders, parasomnias, sleep-related movement disorders, and other sleep disorders [52].

Sleep disorders are common in patients with epilepsy with a prevalence ranging 24–55% [53,54,55,56]. The prevalence of sleep disorders in this group of patients is 2–3 times higher than in the general age-matched population [53,57,58,59]. Especially in patients with drug resistance seizures, problems with sleep are more frequently observed [60,61,62]. Xu et al. showed a higher prevalence of sleep disturbances in women with focal-onset epilepsy than in men [56]. Both seizures and sleep phases of are manifestations of changes in the bioelectrical activity of the brain. The basic diagnostic tool in both phenomena is the analysis of electroencephalographic recordings. The three most common primary sleep disturbances in epilepsy are insomnia, sleep-disordered breathing, and restless legs syndrome (RLS).

## 6. Effect of microRNAs on Primary Sleep Disorders Associated with Epilepsy

### 6.1. Insomnia

The most common sleep problem in patients with epilepsy is insomnia, which has an impact on seizure control and a negative influence on quality of life in patients with epilepsy [52,63,64]. According to questionnaire studies, the prevalence of insomnia in people with epilepsy is estimated at about 52% [57,58]. Yang et al. [65] reported that the severity of insomnia in patients with epilepsy is associated with coexisting depression or another medical condition such as asthma/chronic obstructive pulmonary disease, head trauma, sedative–hypnotic use, and AED polytheraphy [65]. The same results were obtained by Vendrame et al., who reported that insomnia severity was significantly related to the number of anti-epileptic medications and depressive symptoms [63]. Patients with epilepsy and insomnia have frequent daytime somnolence and depressive symptoms. Sleep loss induces changes in many mRNA species. In particular, the levels of microRNA-125a, microRNA-126, and microRNA-146a are significantly lower in short-sleep compared with normal sleep groups. Dysregulated miRNA-146A in patients with fatal familial insomnia causes increased tau hyperphosphorylation [6].

### 6.2. Sleep-Disordered Breathing

The term obstructive sleep apnea (OSA) encompasses a group of disorders characterized by temporary problems (apnea or hypopnea) with breathing during sleep. Several studies have reported a high frequency of OSA in adult patients with epilepsy [54]. The incidence of OSA in patients with epilepsy is estimated at about 20–40% and is more frequent in patients with medical refractory seizures [54,60,61,62]. Some microRNA can be involved in the pathophysiology of OSA. Li et al. [66] demonstrated that hsa-miR-485-5p, hsa-miR-107, hsa-miR-574-5p, and hsa-miR-199-3p might participate in OSA. They observed a difference between OSA patients and a healthy group. CPAP therapy in patients with OSA in the general population is connected with reduction of daytime sleepiness and improvement of cognitive functions and better quality of life [67,67,68,69]. In addition, in epilepsy patients, therapy with CPAP has been associated with significant reduction in seizure frequency and daytime sleepiness compared to untreated patients [70]. MicroRNAs also play an important role in some pathways related to common diseases associated with OSA. Thus, CPAP use associated with changes in the miRNA profile could influence the overall risk of suffering from OSA-related diseases [71] (Table 2).

## 7. Conclusions

MicroRNA expression testing may become a useful diagnostic and predictive tool for differential diagnosis of sleep disorders and sleep-related epilepsy through the development of specific biomarkers.

Reports on the utility of microRNA expression in sleep disorders and sleep-related epilepsy indicate potential for the development of new therapeutic approaches for these disorders by using antisense microRNA (antagomirs) or “repair” drugs designed to compensate for particle abundance when genes for specific microRNAs are deleted.

## Figures and Tables

**Table 1 ijms-22-07370-t001:** Dysregulated miRNAs in the patient with sleep-related epilepsy syndromes [31,32,35,36,37,38,39].

	miRNA	Gene Type
sleep-related hypermotor epilepsy (SHE)	miRNA-211	CHRNA4, CHRNA2 CHRNB2
Panayiotopoulos syndrome	miRNA-155	SCN1A
benign partial epilepsy with centrotemporal spikes (BECTS)	miRNA-328	PAXNEB GRIN2A (atypical form)
idiopathic generalized epilepsy, including juvenile myoclonic epilepsy (JME)	miR 194-5p and miR 106b	EFHC1, ClCN2, KCNQ3, KCNMB3, GABRA1,BRD2, KCNJ10, CACNA1A

**Table 2 ijms-22-07370-t002:** Dysregulated miRNAs in primary sleep disorders in patients with epilepsy, associated with regulation of clock genes [6,7].

Primary Sleep Disorders	miRNA	Rhythmicity	Regulation	Target Clock Gene	Predicted Disease Mechanism
insomnia	miRNA-125a	ND	hippocampus	Per3; CKIε,	long-term regulation of sleep
	miRNA-126	ND	dopaminergic neurons	Dpb	dysregulation of trophic support in DA neurons
	miRNA-146a	rhythmic	frontal cortex, hippocampus	n.d.	increased tau hyperphosphorylation
obstructive sleep apnea	miRNA-107	rhythmic	temporal cortex	CLOCK gene	increased BACE1 expression

n.d. = no data.

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
