# Peer review of "Regulation of microRNA Expression in Sleep Disorders in Patients with Epilepsy"

_ijms, 2021, doi:10.3390/ijms22147370_

Round 1

Reviewer 1 Report

The paper concerns the role of microRNAs in sleep disorders in patients with epilepsy. It is interesting and well-written but the part on the classification of sleep disorders is much too long. The authors should reduce this part and add a new section about an issue that has been completely missed but is of extreme interest in molecular sciences: the role of microRNAs in SUDEP (sudden death in epilepsy). This addition would greatly improve the scientific appeal of the paper. The authors could consider these papers:

  • Sforza et al: Sudden unexpected death in epilepsy: Small RNAs raise expectations Epilepsy Behav 2013 Dec;29(3):591-3.doi: 10.1016/j.yebeh.2013.08.009. Epub 2013 Oct 9.
  • Coll et al: Update on the Genetic Basis of Sudden Unexpected Death in Epilepsy. IJMS 2019 Apr 23;20(8):1979. doi: 10.3390/ijms20081979.
  • Partemi S, et al Genetic and forensic implications in epilepsy and cardiac arrhythmias: a case series. Int J Legal Med. 2015 May;129(3):495-504. doi:
    10.1007/s00414-014-1063-4. Epub 2014 Aug 15. PMID: 25119684.

Author Response

The authors would like to thank the Reviewer for the thorough and insighful comments. We  have tried our best to correct and improve the manuscript according to the Reviewer' remarks.

The relevant data were provided.

The article was supplemented with information on SUDEP, especially miRNA disturbance. The literature has also been supplemented. The section on sleep disorders where no data on miRNA disorders was found has been shortened.

Reviewer 2 Report

The authors majorly have revised the manuscript. This revision is now suggestive of potential utility of abnormal microRNA expression in sleep disorder associated with epilepsy.

Major issues.

#1. I still am not convinced why you chose the epilepsies that were SHE, PS, BECTs and JME ? Why did not you choose Grand mal upon awaking? There are other many epilepsies that are related to sleep.

#2. Tha facts that insomnia, OSA, restless leg, parasomnia, … that you showed in the manuscript are all related to sleep is understandable. However, how about relation sip between microRNA and epilepsy? You cited references that showed higher prevalence of them in patients with epilepsy, but I do not see sleep disorders in patients with epilepsy that you showed in the manuscript is commonly accepted.

Minor issues:

Polysomnography is not important tool to diagnose epilepsy.

Author Response

We  appreciate all the thorough and insightful comments of the Reviewer. We  have tried our best to correct and improve the manuscript according to the Reviewer' remarks.

The relevant information  was provided.

The manuscript was supplemented with other sleep-related epilepsy. The syndromes with disrupted miRNA regulation are discussed.

The part on sleep disorders was shortened.

Information on polysomnography, which is not important in the diagnosis of epilepsy, has been removed.

Round 2

Reviewer 1 Report

the authors fulfilled the requested modifications.

Reviewer 2 Report

I endorse this revision. 

This manuscript is a resubmission of an earlier submission. The following is a list of the peer review reports and author responses from that submission.

Round 1

Reviewer 1 Report

Summery

This is a narrative type of review article. The authors suggest that microRNA regulation is related to both sleep and epileptic disorders.

Major issues.

#1. From epileptological point of views, sleep related epileptic seizures cannot be limited to the three syndromes. Additionally, these syndromes are not specifically known as the sleep related epileptic seizures. Why the authors chose these three? Even though, these three are related to genetical disorders, abnormality in microRNA in the three syndromes are not known.

#2. Do the authors want to use the term epileptic “seizures”? On one side, they use the term seizures, on the other side, they use syndrome. I guess the authors are confused with and could not differentiate these terms.

#3. I am not convinced by the contents in the manuscript because microRNA is not definitively as a biomarker in both epilepsy and sleep disorders. Since this is a narrative review article, the authors should concentrate on reviewed articles about microRNA on sleep and epilepsy. Are all disorders the author described in the paper related to microRNA?

Minor issues

#1. The word “nanapides”  is unclear to me. This might induce confusion for readers. Why the authors want to use this type of terminology is not understandable for me.

#2. Are three key words, ok? Generally, more.

#3. There are many mispunctuations, misspellings, different font size and inconsistent abbreviations . Please care.

I am sorry for not accepting this paper. Generally, I do not reject any papers and try to amend them as much as possible. However, this time, as I pointed out the font size, inconsistent abbreviation etc., I regard this paper as in a pre-submission level. The author is required to make the paper sophisticated for publication first.    

Reviewer 2 Report

The authors propose the review of the available medical database in search of other markers that assess correlations between epilepsy and sleep. Although the subject is very interesting the review is limited and not well documented from the literature point of view.

It is well oriented the final message that learning more about the role of microRNAs in the pathomechanism of sleep disorder epilepsy may result in its use as a biomarker in these disorders and application in therapy, but the review doesn't add any new clues on future research.